cellular biology, ecology, microbiology

*Cassiopea*, photosynthesis, stable isotope labelling, nutrients, electron microscopy, NanoSIMS

**Author for correspondence:**
Niclas Heidelberg Lyndby
e-mail: niclas.lyndby@epfl.ch

# Amoebocytes facilitate efficient carbon and nitrogen assimilation in the *Cassiopea*-Symbiodiniaceae symbiosis

Niclas Heidelberg Lyndby[1], Nils Rädecker[1], Sandrine Bessette[1], Louise Helene Søgaard Jensen[1], Stéphane Escrig[1], Erik Trampe[2], Michael Kühl[2] and Anders Meibom[1,3]

[1]Laboratory for Biological Geochemistry, School of Architecture, Civil and Environmental Engineering, Ecole Polytechnique Fédérale de Lausanne (EPFL), CH-1015 Lausanne, Switzerland
[2]Marine Biological Section, Department of Biology, University of Copenhagen, DK-3000 Helsingør, Denmark
[3]Center for Advanced Surface Analysis, Institute of Earth Sciences, University of Lausanne, CH-1015 Lausanne, Switzerland

(iD) NHL, 0000-0003-0533-9663; ET, 0000-0003-3249-0297; MK, 0000-0002-1792-4790

The upside-down jellyfish *Cassiopea* engages in symbiosis with photosynthetic microalgae that facilitate uptake and recycling of inorganic nutrients. By contrast to most other symbiotic cnidarians, algal endosymbionts in *Cassiopea* are not restricted to the gastroderm but are found in amoebocyte cells within the mesoglea. While symbiont-bearing amoebocytes are highly abundant, their role in nutrient uptake and cycling in *Cassiopea* remains unknown. By combining isotopic labelling experiments with correlated scanning electron microscopy, and Nano-scale secondary ion mass spectrometry (NanoSIMS) imaging, we quantified the anabolic assimilation of inorganic carbon and nitrogen at the subcellular level in juvenile *Cassiopea* medusae bell tissue. Amoebocytes were clustered near the sub-umbrella epidermis and facilitated efficient assimilation of inorganic nutrients. Photosynthetically fixed carbon was efficiently translocated between endosymbionts, amoebocytes and host epidermis at rates similar to or exceeding those observed in corals. The *Cassiopea* holobionts efficiently assimilated ammonium, while no nitrate assimilation was detected, possibly reflecting adaptation to highly dynamic environmental conditions of their natural habitat. The motile amoebocytes allow *Cassiopea* medusae to distribute their endosymbiont population to optimize access to light and nutrients, and transport nutrition between tissue areas. Amoebocytes thus play a vital role for the assimilation and translocation of nutrients in *Cassiopea*, providing an interesting new model for studies of metabolic interactions in photosymbiotic marine organisms.

## 1. Introduction

Animal–microbe symbioses represent a fundamental pillar of life in most habitats [1]. In the marine environment, the symbiotic relationship between cnidaria and endosymbiotic dinoflagellate algae of the family Symbiodiniaceae have been key to the evolutionary success of these animals [2–4], epitomized by reef-building corals. Metabolic interactions between the coral host and endosymbiont dinoflagellate algae, as well as a diverse microbiome (collectively referred to as the coral holobiont, [5–7]), facilitate efficient assimilation and recycling of organic and inorganic nutrients to support the formation and maintenance of entire coral reef ecosystems [8–11]. The translocation of photosynthetically fixed carbon from the algae to the coral host provides the major energy source for the metabolism of the host, which in turn supplies inorganic carbon to the algal photosynthesis from its respiration [9,10,12]. Besides efficient carbon cycling, the coupling of heterotrophic and phototrophic metabolism also enables coral

holobionts to efficiently assimilate and recycle otherwise limiting inorganic nutrients, such as nitrate and ammonium [13]. At the same time, these tight metabolic interactions also render the corals highly vulnerable to the breakdown of the symbiosis (i.e. coral bleaching), which is now reoccurring on a massive scale in many reef localities [14–16]. Whether driven by global warming of seawater or local environmental stress, bleached corals no longer receive the same metabolic input from their diminished dinoflagellate symbiont population and can starve to death on a timescale of days and weeks, causing entire reef ecosystems to collapse [17,18]. Because of the fundamental importance of the cnidarian-dinoflagellate symbiosis to marine life—and urged on by the coral reef crisis [14]—there is a strong effort to identify suitable model organisms that can bring forth novel experimental opportunities and help illuminate the intricate metabolic relationships between cnidarian hosts and their symbiont populations. By contrast with corals, scyphozoans seem to be thriving despite the warming of ocean waters [19]. Often referred to as the 'true jellyfish', this class of cnidaria also host symbiont dinoflagellate algae and one member of the scyphozoans, the upside-down jelly fish *Cassiopea*, has recently received increasing attention as a good model system for studies of host algal metabolic interactions [20].

*Cassiopea* spp. exhibits unique characteristics that set it apart from corals [20]: (i) by contrast to the sessile corals, the life cycle of *Cassiopea* is dominated by a motile medusa stage (i.e. the adult form), which allows the animal to relocate in search for environmentally optimal conditions; (ii) in the medusa stage, *Cassiopea* exhibits the distinct morphological characteristics of scyphozoans [21], with epidermis and gastrodermis spatially separated by a (compared to corals) very large mesoglea [22]; and (iii) the microalgal symbionts in *Cassiopea* are, at least early in the medusa stage, predominantly found inside motile amoebocyte cells located within the mesoglea [23,24]. By contrast, corals host algal endosymbionts exclusively inside their gastrodermal cells.

Previous studies have demonstrated that active nutrient exchange does take place between *Cassiopea* and its algal endosymbiont population [25–27], but the analytical methods employed did not allow disentanglement—at the tissue and single-cell level—the role of algal-containing amoebocytes in nutrient assimilation and the metabolic transfer to adjacent tissue structures. Here, we present a tissue- and (sub)cellular level investigation of autotrophic carbon and nitrogen assimilation and translocation in juvenile *Cassiopea* sp. medusae, in which the endosymbiotic algae are exclusively (i.e. without observed exception) restricted to amoebocytes. By coupling isotopic tracer (i.e. $^{13}C$-bicarbonate, $^{15}N$-ammonium and $^{15}N$-nitrate) incubation experiments with correlated ultrastructural imaging and Nano-scale secondary ion mass spectrometry (NanoSIMS) [28–35], we quantified the anabolic turnover in individual tissues and cellular compartments. This allowed us to disentangle the role of symbiotic amoebocytes for holobiont nutrition and illuminate the ecological advantages of this unique feature of the *Cassiopea*-Symbiodiniaceae symbiosis.

## 2. Material and methods

### (a) *Cassiopea* husbandry
Adult specimens of *Cassiopea* sp. medusa originally obtained from DeJong Marinelife (Netherlands) were cultivated at the Marine Biology Section (MBS), University of Copenhagen (Helsingør, Denmark), that provides a steady supply of all life stages of the jellyfish. Juvenile *Cassiopea* sp. medusae with an umbrella diameter of 6–10 mm were reared in a 20 l aquarium. The jellyfish were fed five times per week with newly hatched *Artemia* nauplii. Animals used for incubations were heterotrophically starved for 24 h before experiments started. All animals were kept at 25°C in artificial seawater (ASW) with a salinity of 35 ppt and a pH of 8.1. Water was recycled inside the tank (20 l) using a small filter pump. Light was maintained with a programmable LED light source (Triton R2, Pacific Sun), running a 12 : 12 h day : night cycle with a downwelling photon irradiance (400–700 nm) of 500 μmol photons $m^{-2} s^{-1}$, measured just above the water surface using a calibrated spectroradiometer (MSC-15, GigaHertz-Optik).

### (b) Experimental setup
A water bath with approximately 10 l of deionized water was placed on top of three magnet stirrers (RCT basic, IKA GmbH). The bath was fitted with a heater to keep the water at 25 (±0.5)°C and a small pump to keep the water well mixed. All incubations were run in triplicate and each *Cassiopea* medusae was incubated in a separate 100 ml beaker filled with 95 ml ASW and positioned in the water bath. Each beaker was fitted with a mesh mounted a few centimetres above the bottom to separate the jellyfish from the magnet bar during incubations. Magnet stirrers were set to spin at 180 rounds per minute (RPM) to ensure mixing of water during incubations. The incubation chambers were illuminated by three identical tungsten halogen lamps (KL 2500 LCD, Schott AG). Each adjustable light source was equipped with a fibre guide and a collimating lens, and the incident photon irradiance for each chamber was adjusted to 350 μmol photons $m^{-2} s^{-1}$ (400–700 nm; electronic supplementary material, figure S1). This light level was chosen as a photosynthetically optimal irradiance (just below saturation point) based on previous measurements of relative electron transport rates versus photon irradiance with a variable chlorophyll fluorescence imaging system (I-PAM, Walz GmbH, Germany) measuring rapid light curves (electronic supplementary material, figure S2; [36]). Photon irradiance measurements were done with a calibrated spectroradiometer (MSC-15, GigaHertz-Optik).

### (c) Isotopic pulse labelling experiments
#### (i) Preparation of isotopically enriched seawater
ASW in the main holding tank at MBS had been monitored for concentrations prior to experiments, and showed a $NO_3^-$ level of 0.01 mg $l^{-1}$, suggesting overall low levels of dissolved nitrogen in the system. Approximately 15 l of this water was collected and stored in a cold room (4°C) until use, a few days later. The evening prior to start of the experiment, 1 l of the collected water was filtered (Millipore, 0.2 μm) and kept overnight at 4°C. The next morning, the filtered ASW was first stripped for dissolved inorganic carbon by reducing the pH to less than 3 by the addition of 1 M HCl and subsequent flushing with ambient air for 2 h. The pH was monitored with a calibrated pH metre (PHM220, MeterLab). Right before start of the experiment, 1 l of $CO_2$-stripped, low pH ASW was isotopically enriched by spiking the water with 255 mg of NaH$^{13}$CO$_3$ (99 atom%, Sigma Aldrich), and 1 ml of stock solutions with either 3 mM K$^{15}$NO$_3$ (99 atom %, Sigma Aldrich) or 3 mM $^{15}$NH$_4$Cl (99 atom%, Sigma Aldrich) in milli-Q water. The pH was then increased by the addition of about 1 ml of 1 M NaOH. The final incubation medium had a pH of 8.0–8.1 and contained 3 mM NaH$^{13}$CO$_3$ and 3 μM of either K$^{15}$NO$_3$ or $^{15}$NH$_4$Cl, with a salinity of 35 ppt. The experimental concentrations of bicarbonate (3 mM) and nitrate or ammonia (3 μM) are high, but not uncommon for waters around

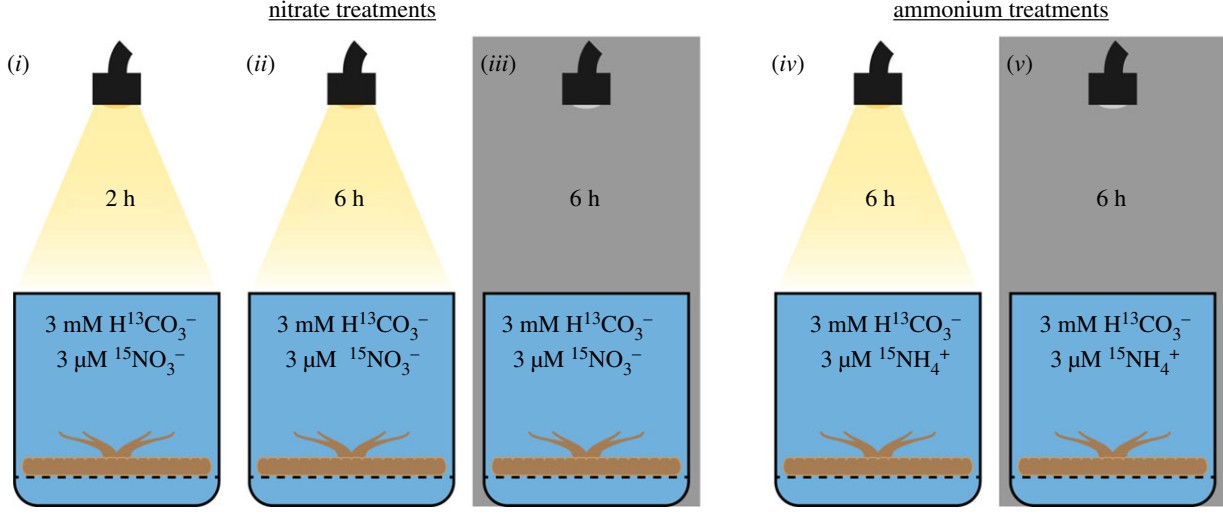

**Figure 1.** Schematic overview of the five different incubations of *Cassiopea* sp. in stable isotope-spiked artificial seawater (ASW). All incubations were run in triplicate, and each individual jellyfish was incubated in 95 ml filtered ASW containing 3 mM $NaH^{13}CO_3$ and either 3 µM $K^{15}NO_3$ (incubations *i*, *ii* and *iii*) or 3 µM $^{15}NH_4Cl$ (incubations *iv* and *v*). All light incubations where run with a light intensity of 350 µmol photons $m^{-2}$ $s^{-1}$, while all dark incubations were done in complete darkness. (Online version in colour.)

Cuba where the animals originate [37,38]. The spiked ASW solution was then heated to 25°C in a water bath and used for incubations.

### (ii) Pulse-label incubations

*Cassiopea* specimens contained in beakers were submitted to one out of five different pulse-label incubations, with a combination of 3 mM $NaH^{13}CO_3$ and 3 µM $K^{15}NO_3$ for (*i*) 2 h in light, (*ii*) 6 h in light, or (*iii*) 6 h darkness, or with 3 mM $NaH^{13}CO_3$ and 3 µM $^{15}NH_4Cl$ for (*iv*) 6 h in light, or (*v*) 6 h in darkness (figure 1). During the 6 h incubations, the solutions were changed at 2 and 4 h.

At the end of incubations, the *Cassiopea* specimens were chemically fixed in cryotubes containing 1.5 ml of 2.5% [v/v] glutaraldehyde (Electron Microscopy Sciences), 0.5% [v/v] paraformaldehyde (Electron Microscopy Sciences) and 0.6 M sucrose (Sigma Aldrich) mixed in 0.1 M Sörensen phosphate buffer. The chemically fixed *Cassiopea* specimens were kept at room temperature for 2 h and were then stored in a fridge at 4°C until transportation to Lausanne (Switzerland) in fixative for further processing.

### (iii) Histological sectioning for electron microscopy imaging and NanoSIMS analyses

Fixed *Cassiopea* specimens were rinsed 3 times in 0.1 M Sörensen buffer with 30 min between changes. Samples were subsequently dissected to acquire a radial slice of the umbrella, from the central manubrium to the bell margin, as illustrated in figure 2*a*,*b*. In order to preserve lipids in the tissue, dissected tissue was submerged in $OsO_4$ solution (2% in milli-Q water) for 30 min under constant slow rotation in darkness. Samples were subsequently rinsed extensively in deionized water (three changes at 30–60 min intervals), before further sample preparation.

Dissected tissue samples underwent dehydration via immersion in a milli-Q water/ethanol series (50, 70, 90 and 100% ethanol), and subsequent resin infiltration with ethanol/Spurr's resin (30, 50, 75 and 100% resin). Resin-infiltrated samples were oriented and placed in moulds filled with 100% Spurr's resin and cured for 72 h in an oven at 60°C. Semi-thin histological sections (500 nm) were cut from the resin blocks using an Ultracut S microtome (Leica Microsystems), equipped with a 45° histo-diamond knife (DiATOME). One to two sections were placed on 10 mm round glass coverslips, coated with a *ca* 12 nm layer of gold using a Leica EM SCD050 gold coater (Leica Camera AG) and mounted in a sample holder for NanoSIMS isotopic imaging.

Additionally, selected thin sections were placed on a microscope slide and stained with Epoxy Tissue Stain (Electron Microscopy Sciences) for 30–60 s on a heating plate; excessive stain was subsequently removed with deionized water, followed by heat-drying. The stained sections were imaged on an optical microscope (Axio Imager.M2 m, ZEISS) to guide NanoSIMS analyses.

### (iv) Scanning electron microscopy imaging

Semi-thin sections (500 nm) were cut as described above, and placed on silicon wafers before being stained with 4% uranyl acetate and Reynolds lead citrate solution (Electron Microscopy Sciences). Images were taken on a GeminiSEM 500 field emission scanning electron microscope (SEM; ZEISS), at 3 kV, an aperture size of 20 µm, and a working distance of 2.5–2.7 mm, using the energy selective backscatter detector (EsB; ZEISS) with the grid set at 382–500 V. Images were acquired directly from the GeminiSEM 500 in TIFF format without further processing.

### (v) NanoSIMS image acquisition

Isotopic imaging of semi-thin histological sections was done with a NanoSIMS 50 L instrument. Images (40 × 40 or 45 × 45 µm, 256 × 256 pixel, 5000 µs pixel$^{-1}$, five layers) were obtained with a 16 KeV $Cs^+$ primary ion source focused to a spot-size of about 120 nm (2 pA). Secondary ions ($^{12}C_2^-$, $^{13}C^{12}C^-$, $^{12}C^{14}N^-$ and $^{12}C^{15}N^-$) were counted in individual electron-multiplier detectors at a mass resolution power of about 9000 (Cameca definition), sufficient to resolve all potential interferences in the mass spectrum.

Isotopic images were analysed using the NanoSIMS software L'IMAGE (v.12–21–2017; developed by Dr Larry Nittler, Carnegie Institution of Washington), and contours were carefully drawn in each image around the epidermis as well as individual amoebocytes and dinoflagellate cells. Epidermis were counted as one region of interest (ROI; $n = 1$) for each image. Similarly, amoebocytes (clustered with dinoflagellates) were usually treated as one ROI per image unless cell clusters were clearly separated. Drift-corrected maps of $^{13}C$- and $^{15}N$ enrichment were obtained from the count ratios $^{13}C^{12}C^-/^{12}C_2^-$ and $^{15}N^{12}C^-/^{14}N^{12}C^-$, respectively. Measured enrichments were expressed in the delta notation:

$$\delta^{13}C \ (‰) = \frac{r_{eC} - r_{cC}}{r_{cC}} \times 1000 \qquad (2.1)$$

and

$$\delta^{15}N \ (‰) = \frac{r_{eN} - r_{cN}}{r_{cN}} \times 1000, \qquad (2.2)$$

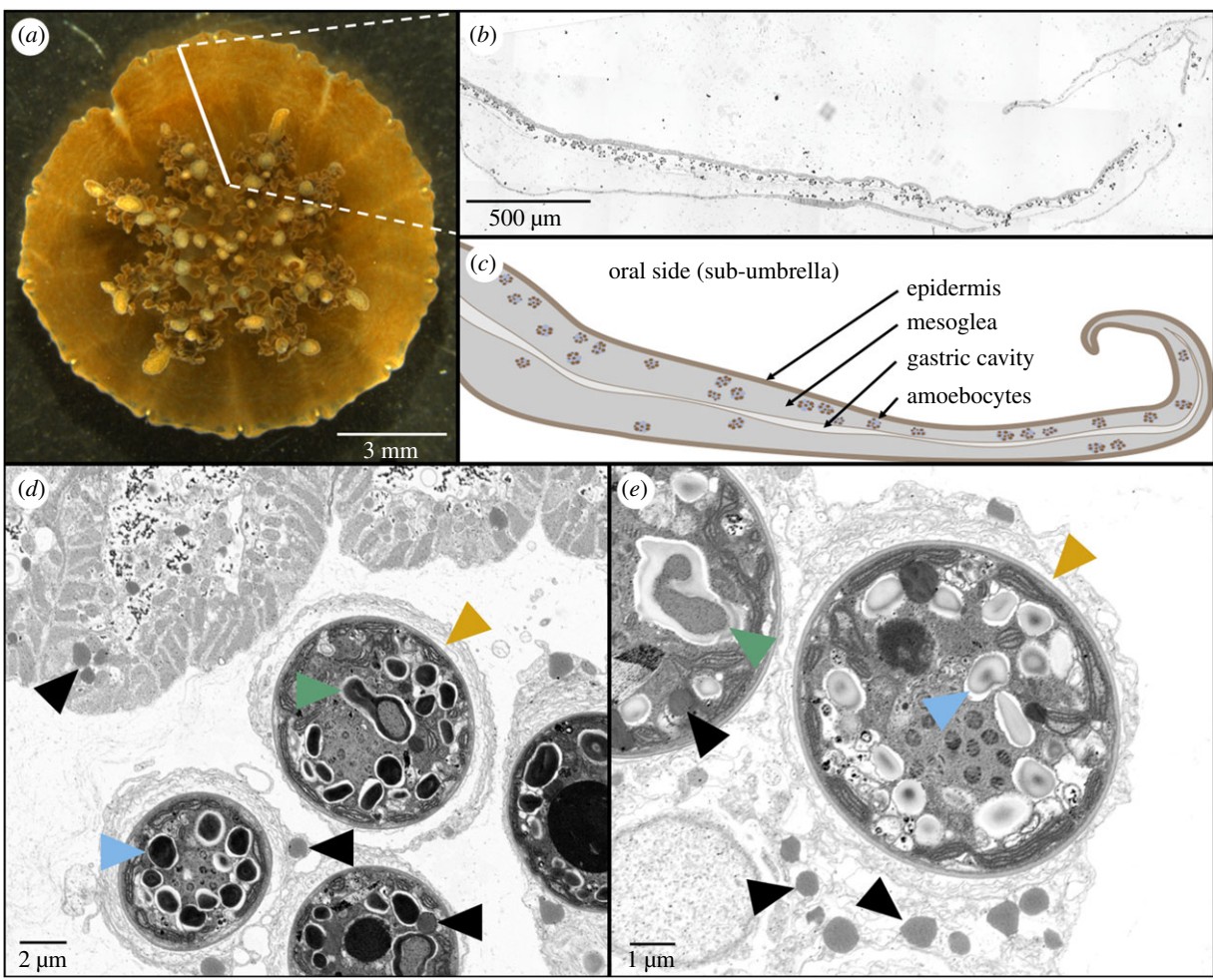

**Figure 2.** Histological and electron microscopy images of *Cassiopea* tissue architecture. (*a*) Photograph of a juvenile *Cassiopea*. The solid white line indicates the orientation of histological sections cut for this study; i.e. cutting radially from the central manubrium region to the edge of bell. (*b*) Optical microscopy image of a histological section as indicated in (*a*). (*c*) Sketch of histological section illustrating key tissue structures. (*d* and *e*) Scanning electron microscopy showing the micro-architecture of host amoebocytes densely populated with symbionts next to the host epidermal tissue. Note that the grey-scale of (*e*) is inverted relative to (*d*) for increased clarity of subcellular components. Subcellular components are indicated: lipid bodies in symbionts (black triangles), pyrenoids (green triangles), and starch granules (blue triangles). Orange triangles indicate symbiosomes (i.e. host vacuolar space surrounded by a membrane similar to what is known from other symbiotic cnidarians hosting dinoflagellates) in amoebocytes. (Online version in colour.)

where $r_{eC}$ and $r_{cC}$ are the count ratios of $^{13}C^{12}C^-/^{12}C_2$ in an enriched sample and a control (i.e. unlabelled) sample, respectively. $r_{eN}$ and $r_{cN}$ are the count ratios of $^{15}N^{12}C^-/^{14}N^{12}C^-$ in an enriched sample and a control sample, respectively. The limit for detection of isotopic enrichment was obtained by analysing unlabelled (i.e. control) samples and determine the standard deviation among these analyses. The detection limit was then defined as three standard deviations above the control isotopic ratio and all tissue areas with isotopic ratios below this limit were considered unlabelled. Total numbers of technical replicates (ROIs) is provided in the electronic supplementary material, table S1. Control samples with natural isotopic composition were analysed at least twice per day to monitor instrumental drift. However, these control isotope ratios were within 20‰ of each other and no corrections were necessary.

### (vi) Dinoflagellate identification
All animals used in this study contained dinoflagellate symbionts of the genus *Symbiodinium* (previously clade A; [2]). Algal symbionts were identified by DNA extraction, polymerase chain reaction amplification with genus/clade-specific primers (adapted after Yamashita *et al.* [39]) and visualization of amplification using gel electrophoresis (see the electronic supplementary material).

### (d) Statistical analyses
We used R (v.4.0.2) with the packages *nlme* (v.3.1–148) and *lsmeans* (v.2.30–0) to perform statistical analyses. Linear mixed model (LMM) analyses were used to test the relationship between isotopic enrichment in *Cassiopea* sp. holobiont compartments and the conditions of time, light availability and nitrogen source taking into account the biological replicate as a random factor ($n = 3$). $^{13}C$ enrichment data were square root transformed to achieve normality. Tukey *post hoc* analyses were used for tests with more than two groups involved. See the electronic supplementary material, table S1 for the total number of technical replicates per tissue area per treatment.

## 3. Results

### (a) *Cassiopea* histology
The juvenile specimens of *Cassiopea* sp. used in this study showed a high density of dinoflagellate symbionts in the sub-umbrella. All dinoflagellates were found inside amoebocytes clustered in the mesoglea, the vast majority of which were in close proximity to the (sub-umbrella) epidermal tissue layer (figure 2*b*), as previously described [23,24]. Dinoflagellates were never observed outside amoebocytes

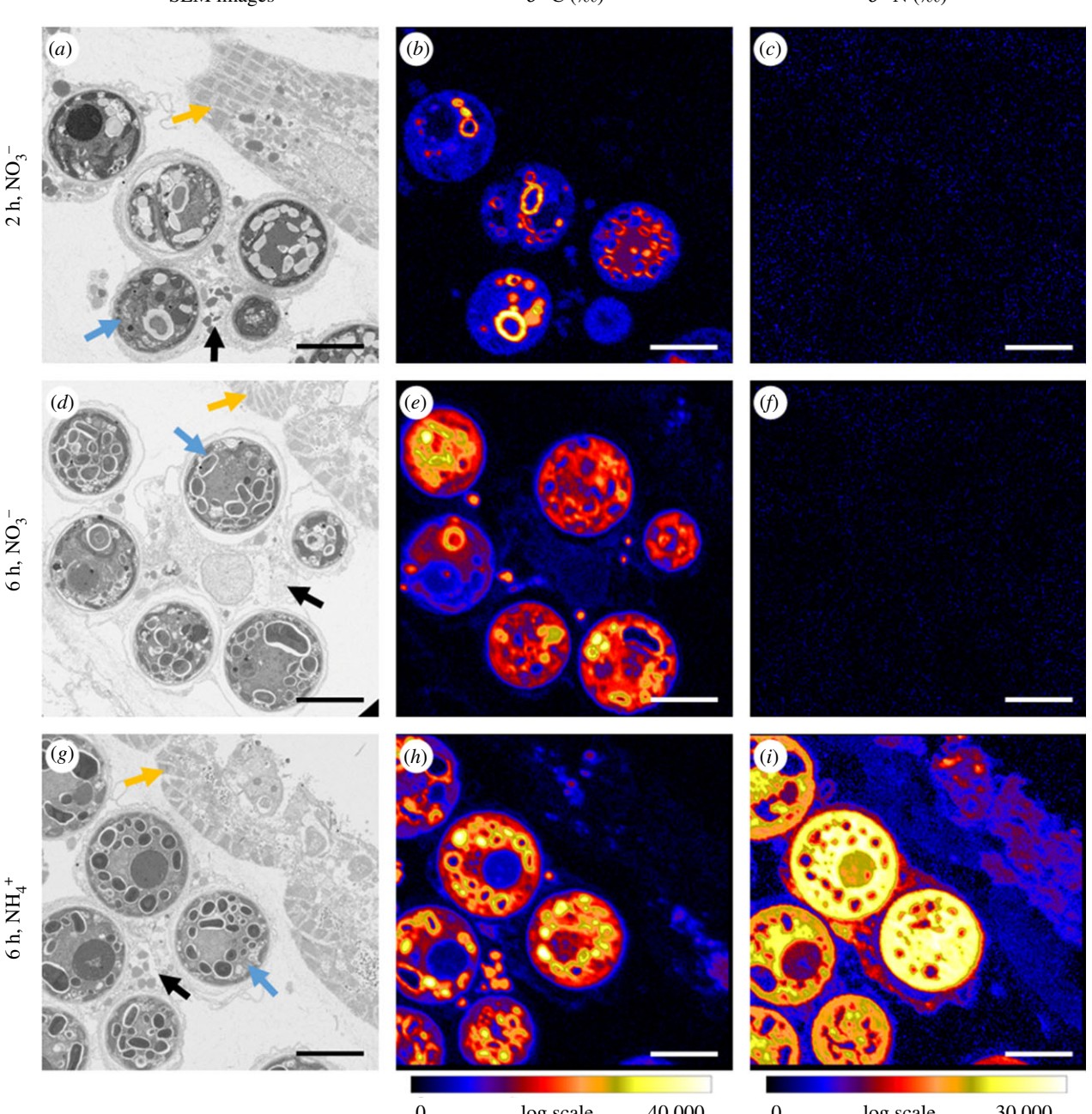

**Figure 3.** Correlated SEM and NanoSIMS images for light incubations of representative tissue areas of *Cassiopea* sp. incubated with $H^{13}CO_3^-$ and $^{15}NO_3^-$ for 2 h (*a–c*) and 6 h (*d–f*), respectively, and with $H^{13}CO_3^-$ and $^{15}NH_4^+$ for 6 h (*g–i*). Yellow arrows in (*a*), (*d*), (*g*) indicate epidermis, black arrows amoebocytes, and blue arrows symbiont cells contained in amoebocytes. Scale bars = 8 μm. (Online version in colour.)

or in other types of host cells or tissue layers in our juvenile specimens. Cross-sections of amoebocytes showed that dino-flagellate cells were contained in a symbiosome-like structure (orange triangles in figure 2*d*,*e*; cf. [40]). Furthermore, high magnification electron microscopy images of amoebocytes/symbionts showed the presence of lipid droplets inside both symbiont and amoebocyte cells (black triangles in figure 2*d*,*e*).

### (b) Stable isotope enrichment in *Cassiopea* sp.

NanoSIMS images were acquired of compact clusters of dinoflagellate symbionts inside amoebocyte cells residing in the mesoglea, particularly in the sub-umbrella tissue area (figure 2; a minimum of 25 dinoflagellate cells were imaged per biological sample, $n = 3$), including representative regions of the (sub-umbrella) epidermal tissue (figure 3). Note that,

because of the classical sample preparation methods employed here, which involves ethanol dehydration and resin embedding, most soluble compounds are lost from the tissue and only structural components, i.e. proteins, fatty acids, RNA, DNA, etc. representing the products of anabolic metabolism, remain [41,42]. Isotopic enrichments shown in the following thus represent the relative anabolic turnover of the tissue during the incubation experiments.

### (c) $^{13}$C-assimilation in the holobiont

The incubation experiments described above resulted in clear $^{13}$C enrichment differences between holobiont compartment, incubation time and light availability (electronic supplementary material, table S1). As expected, *Cassiopea* sp. incubated

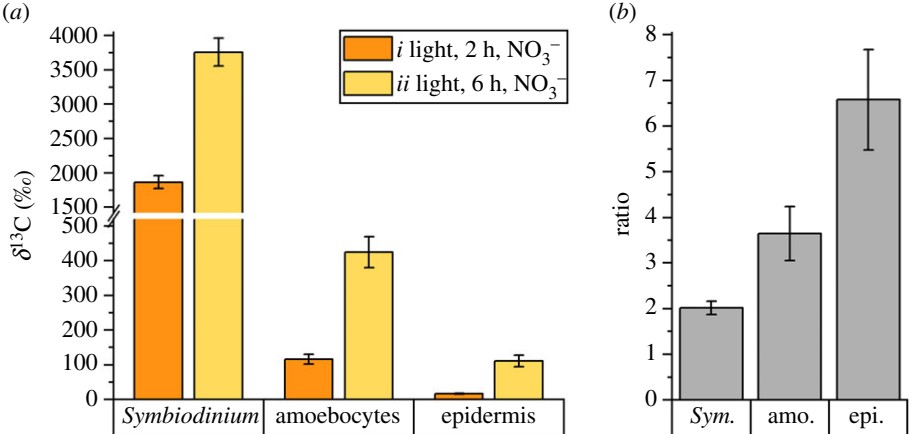

**Figure 4.** (a) Observed mean $^{13}$C enrichment levels in *Symbiodinium* cells and *Cassiopea* sp. host tissue for specimens incubated with $^{13}$C-bicarbonate (and $^{15}$N-nitrate, treatments *i* and *ii*) for 2 and 6 h. Error bars indicate ± s.e. of mean for regions of interests collected across three individual *Cassiopea* medusae per treatment. (b) Fold-change increase in $^{13}$C enrichment for individual tissue areas between 2 and 6 h. Error bars indicate ± propagated errors. (Online version in colour.)

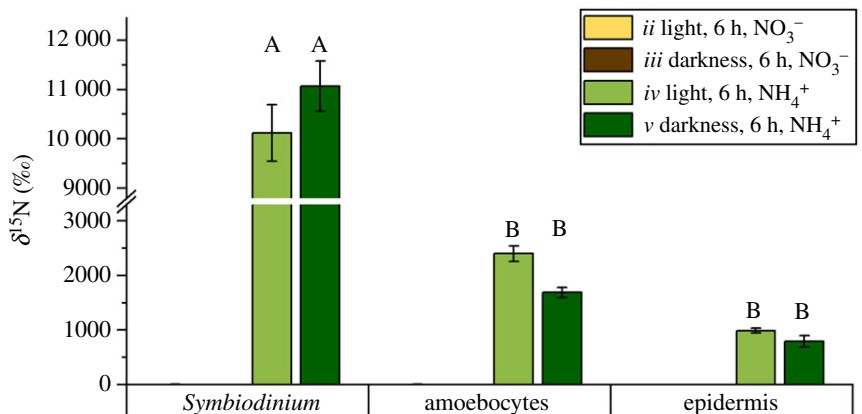

**Figure 5.** Observed $^{15}$N enrichment levels in dinoflagellate symbionts and *Cassiopea* sp. host tissue for specimens incubated with $^{15}$N-nitrate (treatments *ii* and *iii*) and $^{15}$N-ammonium (treatments *iv* and *v*) 6 h in light (treatments *ii* and *iv*) and 6 h in darkness (treatments *iii* and *v*). Error bars indicate ± s.e. of mean for regions of interests collected across three individual *Cassiopea* medusae per treatment. Capital letters indicate significant differences ( $p < 0.05$; Tukey *post hoc* analyses). See the electronic supplementary material, figure S3 for detailed data distribution. (Online version in colour.)

with H$^{13}$CO$_3^-$ in darkness did not show detectable levels of $^{13}$C enrichment (electronic supplementary material, table S1).

By contrast, after 2 and 6 h light incubations, the tissue/cell types (*Symbiodinium* versus amoebocytes versus epidermis) showed characteristic differences in $^{13}$C enrichment with the dinoflagellates showing the highest and the epidermis the lowest levels of enrichment (LMM, $F_{1,219} = 280.7$, $p < 0.001$; figure 4a). The $^{13}$C enrichment increased in all three tissue/cell types in the 6 h compared to the 2 h light incubation, but the relative increase was different for each compartment (LMM, $F_{2,219} = 3.9$, $p = 0.021$; figure 4b). Specifically, *Symbiodinium* $^{13}$C enrichment increased 2-fold (albeit differences were not significant: LMM, $F_{1,4} = 3.70$, $p = 0.127$), amoebocyte $^{13}$C enrichment increased 3.6-fold (LMM, $F_{1,4} = 12.9$, $p = 0.023$), and epidermis $^{13}$C enrichment increased 6.5-fold (LMM, $F_{1,4} = 18.0$, $p = 0.013$) between 2 and 6 h, respectively (figure 4b).

At the same time, we observed that $^{13}$C enrichments were not affected by the source of inorganic nitrogen available in the seawater. The 6 h incubations with either $^{15}$NO$_3$ or $^{15}$NH$_4$ showed no significant differences in $^{13}$C enrichment across holobiont compartments (LMM, $F_{1,4} = 0.8$, $p = 0.425$; electronic supplementary material, table S1).

### (d) $^{15}$N-assimilation in the holobiont

Regardless of incubation time and light conditions, medusae incubated with $^{15}$NO$_3^-$ did not exhibit any detectable $^{15}$N enrichments in any of the holobiont compartments (figure 3c,f and figure 5, electronic supplementary material, table S1). By contrast, 6 h incubation with $^{15}$NH$_4^+$ resulted in clear $^{15}$N enrichments during both light and dark incubations for all holobiont compartments. Reflecting observed patterns of $^{13}$C enrichments, the different tissue- and cell types showed characteristic systematic $^{15}$N enrichments, with *Symbiodinium* exhibiting the highest and the epidermis the lowest $^{15}$N enrichment levels during dark as well as light incubations, respectively (LLM, $F_{2,249} = 158.9$, $p < 0.001$); see figure 5 for relevant Tukey post hoc analyses. Light availability made no difference on ammonium assimilation, with $^{15}$N enrichments not significantly different between light and dark incubations across holobiont compartments (LMM, $F_{1,4} = 0.1$, $p = 0.818$).

## 4. Discussion

By contrast to corals and sea anemones that host their endosymbionts in the gastroderm tissue layer [43], the symbionts

of juvenile *Cassiopea* sp. are predominantly found residing in amoebocyte host cells located within the mesoglea. Metabolic interactions and translocation pathways in *Cassiopea* sp. could thus be expected to show both similarities with, and differences from those observed previously in hermatypic corals. Our findings corroborated previous reports of active nutrient exchange between *Cassiopea* and its algal symbionts. At the same time, the high resolution of quantitative NanoSIMS isotopic imaging allowed us to trace the fate of assimilated nutrients in the symbiosis, and quantify their anabolic partitioning across cells and tissues. We clearly observed that motile amoebocytes facilitated efficient assimilation of inorganic nutrients from seawater, and made isotopically enriched nutrients available for anabolic processes in other tissue layers, strongly suggesting that amoebocytes represent a key adaptation of scyphozoan jellyfish to a symbiotic lifestyle.

## (a) Translocation of photosynthetically assimilated $^{13}$C

The anabolic turnover of photosynthetically fixed carbon in *Cassiopea* sp. was found to display nonlinear behaviour as a function of time in all investigated holobiont compartments. The enrichment level of any cell or tissue at any given time point reflects the balance between anabolic incorporation and catabolic consumption of isotopically enriched compounds. As cells become more and more enriched over time, the catabolic consumption of isotopically enriched compounds increases. Consequently, the increase in enrichment over time is expected to asymptotically approach the saturation point (i.e. 100% $^{13}$C enrichment). Such nonlinearity was observed in *Symbiodinium* cells in which, over a 3-fold increase in incubation time (from 2 to 6 h), only a 2-fold increase in $^{13}$C enrichment was observed (from *ca* 1860 to 3760‰; figure 4). Similarly, a nonlinear trend was observed in the amoebocytes and epidermis, which showed a roughly 3.5- and 6.5-fold increase in $^{13}$C enrichment (from *ca* 120‰ to 420‰ in amoebocytes, and from *ca* 20‰ to 110‰ in epidermis), respectively, between the 2 and 6 h light incubations (figure 4). This implies that the rate of anabolic incorporation of isotopically enriched compounds must have accelerated over time in the host tissue. The relative increase in the $^{13}$C enrichment levels over time, thus, reflects the delay at which photosynthetically fixed $^{13}$C became available to the respective cell/tissue. The increasing ratios between 2 and 6 h incubations allow disentanglement of the order of translocation of photosynthetically fixed carbon in the *Cassiopea*-Symbiodiniaceae symbiosis: Symbiodiniaceae translocate photosynthates to hosting amoebocytes, which subsequently can release labelled carbon substrates to other compartments of the holobiont, such as the epidermis.

Importantly, the efficiency of carbon translocation in the *Cassiopea*-Symbiodiniaceae symbiosis appears comparable to that of the coral-Symbiodiniaceae symbiosis. Using a similar methodological approach, Kopp *et al.* [10] observed a qualitatively similar, nonlinear $^{13}$C enrichment curve for storage structures in dinoflagellates hosted by the hermatypic coral *Pocillopora damicornis* during a 6 h pulse labelling with H$^{13}$CO$_3^-$. However, that study found slightly less efficient translocation dynamics, i.e. $^{13}$C enrichment in adjacent host tissue was lower than the $^{13}$C enrichment in the *Cassiopea* amoebocytes and similar to epidermal tissue in the present study. To the degree that such qualitative comparison is possible (i.e. considering differences in the experimental setup,

dinoflagellate species, distribution and density) this—at least qualitatively—indicates that translocation of $^{13}$C-enriched photosynthates is certainly not less efficient in *Cassiopea* compared to that in a symbiotic coral. This conclusion is perhaps surprising considering that symbiotic corals rely on up to 95% of their metabolic energy demand being supplied in the form of photosynthates translocated from their dinoflagellate symbionts [8,10,44]. However, our observations are consistent with previous studies on metabolic interactions between symbiotic jellyfish (such as *Cassiopea*) and their symbiont population, which indicate that these jellyfish can produce more than 100% of their daily carbon demand via autotrophic carbon assimilation [45,46].

Our results thus suggest a very efficient and high-throughput system for autotrophic carbon assimilation in *Cassiopea*, in which a state of saturation of internal storage structures in symbionts is quickly reached and assimilated carbon is rapidly translocated (via amoebocytes) to the host for downstream biosynthesis (figure 4). While we conclude that amoebocytes play a major role in the highly efficient nutrient translocation in the *Cassiopea* holobiont, the exact pathways and underlying mechanisms of such a system in *Cassiopea*, with the added complexity of photosynthates passing from the symbionts, via the amoebocytes, through the mesoglea to the epidermis, remain to be explored.

## (b) Nitrogen assimilation and resource allocation

Patterns of $^{15}$NH$_4^+$ assimilation in the *Cassiopea* holobiont mirrored those previously described in corals [47]. $^{15}$NH$_4^+$ was efficiently assimilated by both host and symbionts, regardless of light availability (figure 5). As both algal endosymbionts and the cnidarian host have the cellular machinery for the metabolic incorporation of NH$_4^+$ [48,49], the observed $^{15}$N enrichment patterns reflect a combination between the cellular incorporation of $^{15}$NH$_4^+$ and the translocation of fixed $^{15}$N between the symbiotic partners. Despite the spatial separation of amoebocytes from the surrounding seawater, Symbiodiniaceae showed the highest $^{15}$N enrichment levels in the *Cassiopea* holobiont. This implies that the close proximity of amoebocytes to the animals' external surface facilitates the, presumably, diffusion-controlled transport of NH$_4^+$ to the algal symbionts.

By contrast to this, $^{15}$NO$_3^-$ assimilation was not observed (or was below the detection limit of the NanoSIMS, estimated to about 20‰ above the natural isotope ratio) regardless of light availability (figure 5). In line with this, Welsh *et al.* [26] found that *Cassiopea* sp. collected from southeast Queensland, Australia, showed only a low net NO$_3^-$ + NO$_2$ (NO$_x$) uptake in the light but attributed a high release of NO$_x$ during dark incubations to the presence of surface-associated nitrifying bacteria. Likewise, Freeman *et al.* [25] found that $^{15}$NO$_3^-$ assimilation in *Cassiopea xamachana* was highest in body parts with the lowest Symbiodiniaceae density. In the light of these studies, our findings suggest that Symbiodiniaceae associated with *Cassiopea* either have limited access to NO$_3^-$ from seawater or exhibit downregulated pathways for NO$_3^-$ uptake [49,50]. In this context, Grover *et al.* [51] previously showed that reduced uptake of NO$_3^-$ by algal symbionts can occur in the presence of elevated NH$_4^+$ concentrations. Hence, the absence of NO$_3^-$ assimilation in the present study could be the result of high *in hospite* NH$_4^+$ availability for algal symbionts owing to the excretion of waste

products by the metabolism of the cnidarian host. Reduced or absent $NO_3^-$ uptake by algal symbionts could indirectly benefit the cnidarian host, as previous studies have linked high rates of $NO_3^-$ assimilation to reduced rates of carbon translocation from algal symbionts, and increased bleaching susceptibility of corals during heat stress [52,53]. Given the wide tolerance to temperatures and salinity that *Cassiopea* exhibits in its natural environment [54,55] (unlike what is commonly known for corals), we hypothesize that a reduced or absent assimilation of $NO_3^-$ by endosymbiotic Symbiodiniaceae may increase stress tolerance of the *Cassiopea* holobiont by enhancing energy availability for the host and reducing the bleaching susceptibility of the symbiosis.

## (c) The ecological advantages of motile symbiont-bearing amoebocytes and *Cassiopea* as a model system

Amoebocytes have not been widely studied in cnidarians [56,57], but they are considered to be beneficial to non-symbiotic jellyfish by acting as part of the immune system [56,58], and take part in general tissue maintenance [59,60]. Amoebocytes in *Cassiopea* have additionally been linked to the establishment of dinoflagellate symbiont populations via phagocytosis in the polyp stage [40,61]. The results of the present study further highlight the role of amoebocytes in the *Cassiopea* holobiont as a key adaptation to facilitate efficient symbiotic nutrient cycling in medusae.

In symbiotic corals, endosymbiotic dinoflagellates are found exclusively in the gastrodermis, which is in direct contact with the gastrovascular cavity and in close proximity to surrounding seawater, from which it is only separated by a thin mesoglea and epidermal tissue layer. By contrast, the thick mesoglea that makes up the bulk part of the medusa constitutes a significant spatial barrier separating gastrodermis from the epidermis and may hinder efficient uptake of nutrients from the seawater by algal symbionts through the gastrodermis. Motile, symbiont-bearing amoebocytes may help *Cassiopea* holobionts to overcome these barriers to nutrient uptake created by the scyphozoan morphology. On one hand, amoebocytes facilitate the relocation of algal symbionts from the gastroderm to other parts in the animal with close proximity to the surface, and better access to nutrients from the seawater. On the other hand, the motile nature of amoebocytes also allow for the release of autotrophically assimilated nutrients in the vicinity of other tissue layers thereby ensuring a more efficient exchange of nutrients in the holobiont. Symbiotic amoebocytes may thus be considered 'cargo-ships', efficiently shuttling symbionts to (photosynthetically and nutrient-wise) optimal locations inside the thick mesoglea, as well as efficiently transporting assimilated nutrients back to host tissue.

Taken together, symbiotic amoebocytes probably reflect a key adaptation of symbiotic scyphozoans to facilitate efficient assimilation and exchange of nutrients. In addition to this, a reduced or absent $NO_3^-$ uptake of algal symbionts may further allow *Cassiopea* to thrive in highly dynamic habitats with strong fluctuations in salinity, nutrient availability and temperature conditions. At the same time, its unique characteristics set the *Cassiopea*-Symbiodiniaceae symbiosis apart from most other cnidarian-algal symbiotic assemblages. While corals and many other symbiotic cnidarians are in global decline owing to anthropogenic environmental change, *Cassiopea* spp. appear to thrive, producing massive blooms and undergoing rapid geographical expansion in the Anthropocene [20,62]. Future studies harnessing the unique advantages and characteristics of the *Cassiopea*-Symbiodiniaceae model system may reveal key processes underlying the functioning and breakdown of cnidarian-algal symbioses in general.

Data accessibility. Raw data are publicly available via the Dryad Digital Repository (https://doi.org/10.5061/dryad.z612jm69h) [63].

Authors' contributions. N.H.L., E.T., M.K. and A.M. designed the experiment. NanoSIMS data acquisition was carried out by N.H.L., N.R., S.E. and A.M.. SEM imaging was carried out by N.H.L. and L.H.S.J. Molecular analyses were carried out by S.B. Data and statistical analyses were carried out by N.H.L. and N.R. All authors contributed to writing and editing the manuscript.

Competing interests. The authors declare no competing interests.

Funding. This study was supported by grants from the Swiss National Science Foundation (A.M.; grant no. 200021_179092), and an Investigator Award from the Gordon and Betty Moore Foundation (M.K.; grant no. GBMF9206, https://doi.org/10.37807/GBMF9206).

Acknowledgements. We thank Sofie Lindegaard Jakobsen at the Marine Biology Section, University of Copenhagen for her excellent assistance with maintenance and care of the *Cassiopea* culture for the duration of this study, and for assistance with equipment for experimental setups. The authors thank Thomas Krueger and Margaret Caitlyn Murray, who performed initial NanoSIMS pilot experiments upon which this study builds. We thank the Electron Microscopy Facility at the University of Lausanne for use of their facilities and expert advice about histological sample preparation. Finally, we thank the two anonymous reviewers for insightful comments that helped improve the manuscript.

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
