## [Reviewer comments · Proceedings of the Royal Society B: Biological Sciences]

Review History

RSPB-2020-2393.R0 (Original submission)

Review form: Reviewer 1

Recommendation

Accept with minor revision (please list in comments)

Scientific importance: Is the manuscript an original and important contribution to its field?

Good

General interest: Is the paper of sufficient general interest?

Good

Quality of the paper: Is the overall quality of the paper suitable?

Good

Is the length of the paper justified?

Yes

Should the paper be seen by a specialist statistical reviewer?

No

Do you have any concerns about statistical analyses in this paper? If so, please specify them explicitly in your report.

No

It is a condition of publication that authors make their supporting data, code and materials available - either as supplementary material or hosted in an external repository. Please rate, if applicable, the supporting data on the following criteria.

Is it accessible?

Yes

Is it clear?

Yes

Is it adequate?

Yes

Do you have any ethical concerns with this paper?

No

Comments to the Author

General comments

This paper was really enjoyable to read. This is an interesting and original study investigating the use of dissolved inorganic nitrogen compounds (NO₃⁻ and NH₄⁺) by jellyfish of the genus *Cassiopea* sp. living in symbiosis with dinoflagellates (*Symbiodinium*, formerly clade A). In comparison with other studies published in the literature so far, this one investigates the incorporation of these two compounds at the tissue and single cell level, using the NanoSIMS imaging technique. The methodology is well explained, and the results obtained are convincing. For these reasons, I recommend publication of the manuscript with minor corrections/clarifications.

Methods

Page 6, « Pulse-label incubations » How were the two incubation periods (2h and 6h) chosen? Did the authors perform preliminary tests with different incubation times or do 2h and 6h represent a kind of "standard incubation time"?

Page 6, « Pulse-label incubations » Has the ammonium concentration been verified in the sea water in which the jellyfish were preconditioned before incubation in the presence of 15N-nitrate? Jellyfish could indeed be expected to excrete ammonium, and this would inhibit the incorporation of nitrates (as also suggested by the authors page 18, lines 9-10).

Page 6, Figure 1 : the light intensity used for incubation could be mention here

Results

Page 12, Figure 3: If possible, it would be useful for a non-specialist reader if the authors emphasize the enrichment in symbionts, amoebocytes and epidermis using arrows (as in Figure 1).

Page 13, line 7: the authors found that "specifically, *Symbiodinium* 13C enrichment increased 2-fold". However the LMM gave non-significant p-value at the 95% CI level in this case, is it correct (p=0,127).

Page 14, lines 4-6: "Regardless of incubation time and light conditions, medusae incubated with 15NO₃⁻ did not exhibit any detectable 15N enrichments in any of the holobiont compartments"

See previous remark (Methods): See previous remark (Methods): Since ammonium is the main form of inorganic N excreted by jellyfish, this could inhibit the absorption of nitrate. This issue could be slightly developed in the Discussion. For example, in scleratinian corals, nitrate uptake rates are found to be significantly lower in presence of high NH_4^+ concentrations in seawater (e.g. Grover et al., *Limnol & Oceanogr.* 2003).

Page 14, Figure 5: The non-overlapping error bars (indicating \pm SE) could be a bit confusing, because the authors found no significant differences between treatments with or without light (possibly add/overlay measurement points on each bar ?)

Page 17, lines 3-4: Note that McCloskey et al. (*Mar Biol.*, 1994) also found that carbon fixed photosynthetically by symbionts satisfy up to 100% of the host's (*Mastigias papua*) daily metabolic carbon demand?

END

Review form: Reviewer 2

Recommendation

Accept with minor revision (please list in comments)

Scientific importance: Is the manuscript an original and important contribution to its field?

Good

General interest: Is the paper of sufficient general interest?

Good

Quality of the paper: Is the overall quality of the paper suitable?

Good

Is the length of the paper justified?

Yes

Should the paper be seen by a specialist statistical reviewer?

Yes

Do you have any concerns about statistical analyses in this paper? If so, please specify them explicitly in your report.

Yes

It is a condition of publication that authors make their supporting data, code and materials available - either as supplementary material or hosted in an external repository. Please rate, if applicable, the supporting data on the following criteria.

Is it accessible?

Yes

Is it clear?

Yes

Is it adequate?

Yes

Do you have any ethical concerns with this paper?

No

Comments to the Author

The authors provide novel insights into the symbiosis formed between the 'upside down jellyfish' (*Cassiopea*) and *Symbiodinium* sp. – a potential model system analogous to corals. Interestingly, in contrast to the recent 'coral crisis', *Cassiopea* have thrived, suggesting comparison of the symbioses formed by the related organisms could lead to important applied outcomes. The elegant and well thought out experiment presented here represents the evolution from previous bulk scale isotope tracer studies on the same organisms (i.e., Freeman et al. 2016). I found the paper well written, easy to follow and likely of reasonably broad interest. I only have a few issues with the paper as is;

- 1) I feel it could be improved with a shortening of paragraphs 2-6 in the discussion. For example, P19 L1-18 could be far more succinct at making the authors point (which I think the authors are overly cautious about).
- 2) I find the handling of replication difficult to follow. The experiment had 5 treatments with 3 replicates of each, yet it seems like replicates were pooled for data analysis and that 'replication' terminology actually refers to selected ROIs? Replication is often a problem with nanoSIMS based studies, but it needs to be clearly outlined as ROIs and true replicates are two very different things. That said, the trends they observe are quite real and their findings are certainly justified.

Few specific comments:

P4 L22-23 – replace 'ultra-high resolution stable isotopic mapping' with high resolution secondary ion mass spectrometry

P6 L23 – how realistic are the isotope spikes? Do these values reflect natural conditions?

P9 L14 – did $^{13}\text{C}_2$ create any interference problems for $^{12}\text{C}^{14}\text{N}$?

P9 L23 – how often was a control sample analysed? Presumably a control sample was analysed periodically through the total analysis time to account for instrument fractionation?

P19 L2 – I don't understand the authors description of 0.2% enrichment detection limit. How do the treatments compare to the control samples? It seems more a question of whether enough data was collected for individual ROIs and how error is then considered. I don't think the authors need to be so cautious here if they are confident with their method.

Decision letter (RSPB-2020-2393.R0)

12-Nov-2020

Dear Mr Lyndby

I am pleased to inform you that your manuscript RSPB-2020-2393 entitled "Amoebocytes facilitate efficient carbon and nitrogen assimilation in the *Cassiopea*-Symbiodiniaceae symbiosis" has been accepted for publication in Proceedings B. Congratulations!!

The referee(s) have recommended publication, but also suggest some minor revisions to your manuscript. Therefore, I invite you to respond to the referee(s)' comments and revise your manuscript. Because the schedule for publication is very tight, it is a condition of publication that

you submit the revised version of your manuscript within 7 days. If you do not think you will be able to meet this date please let us know.

It is a condition of publication that data supporting your paper are made available either in the electronic supplementary material or through an appropriate repository. Please see our Data Sharing Policies <https://royalsociety.org/journals/authors/author-guidelines/#data>.

[http://datadryad.org/submit?journalID=RSPB&manu=\(Document not available\)](http://datadryad.org/submit?journalID=RSPB&manu=(Document+not+available)) which will take you to your unique entry in the Dryad repository. If you have already submitted your data to dryad you can make any necessary revisions to your dataset by following the above link. Please see <https://royalsociety.org/journals/ethics-policies/data-sharing-mining/> for more details.

Sincerely,

Dr John Hutchinson, Editor

Associate Editor

Board Member: 1

Comments to Author:

I am pleased to inform you that your manuscript was reviewed favorably by both referees. We think your manuscript would make a fine contribution to Proceedings B. However, before your manuscript can be accepted for publication, there are still some minor revisions that need to be addressed. Each of the comments made by the reviewers should be addressed before the manuscript can be accepted for publication. In particular, Reviewer 2 seeks better explanation of the replication in the study, and both reviewers have requested clarification of certain items in the Methods and Results. I expect you will find these edits straightforward to make and that the manuscript will benefit from these expert reviews. This revision will not be sent out for further review but will be checked by our Editorial staff to make sure that the paper is suitable to be sent to Production.

Reviewer(s)' Comments to Author:

Referee: 1

Comments to the Author(s)

General comments

This paper was really enjoyable to read. This is an interesting and original study investigating the use of dissolved inorganic nitrogen compounds (NO₃⁻ and NH₄⁺) by jellyfish of the genus *Cassiopea* sp. living in symbiosis with dinoflagellates (*Symbiodinium*, formerly clade A). In comparison with other studies published in the literature so far, this one investigates the incorporation of these two compounds at the tissue and single cell level, using the NanoSIMS imaging technique. The methodology is well explained, and the results obtained are convincing. For these reasons, I recommend publication of the manuscript with minor corrections/clarifications.

Methods

Page 6, « Pulse-label incubations » How were the two incubation periods (2h and 6h) chosen? Did the authors perform preliminary tests with different incubation times or do 2h and 6h represent a kind of "standard incubation time"?

Page 6, « Pulse-label incubations » Has the ammonium concentration been verified in the sea water in which the jellyfish were preconditioned before incubation in the presence of ^{15}N -nitrate ? Jellyfish could indeed be expected to excrete ammonium, and this would inhibit the incorporation of nitrates (as also suggested by the authors page 18, lines 9-10).

Page 6, Figure 1 : the light intensity used for incubation could be mention here

Results

Page 12, Figure 3: If possible, it would be useful for a non-specialist reader if the authors emphasize the enrichment in symbionts, amoebocytes and epidermis using arrows (as in Figure 1).

Page 13, line 7: the authors found that “specifically, Symbiodinium ^{13}C enrichment increased 2-fold”. However the LMM gave non-significant p-value at the 95% CI level in this case, is it correct (p=0,127).

Page 14, lines 4-6: “Regardless of incubation time and light conditions, medusae incubated with $^{15}\text{NO}_3^-$ did not exhibit any detectable ^{15}N enrichments in any of the holobiont compartments” See previous remark (Methods): See previous remark (Methods): Since ammonium is the main form of inorganic N excreted by jellyfish, this could inhibit the absorption of nitrate. This issue could be slightly developed in the Discussion. For example, in scleractinian corals, nitrate uptake rates are found to be significantly lower in presence of high NH_4^+ concentrations in seawater (e.g. Grover et al., *Limnol & Oceanogr.* 2003).

Page 14, Figure 5: The non-overlapping error bars (indicating \pm SE) could be a bit confusing, because the authors found no significant differences between treatments with or without light (possibly add/overlay measurement points on each bar ?)

Page 17, lines 3-4: Note that McCloskey et al. (*Mar Biol.*, 1994) also found that carbon fixed photosynthetically by symbionts satisfy up to 100% of the host's (*Mastigias papua*) daily metabolic carbon demand?

END

Referee: 2

Comments to the Author(s)

The authors provide novel insights into the symbiosis formed between the ‘upside down jellyfish’ (*Cassiopea*) and *Symbiodinium* sp. – a potential model system analogous to corals. Interestingly, in contrast to the recent ‘coral crisis’, *Cassiopea* have thrived, suggesting comparison of the symbioses formed by the related organisms could lead to important applied outcomes. The elegant and well thought out experiment presented here represents the evolution from previous bulk scale isotope tracer studies on the same organisms (i.e., Freeman et al. 2016). I found the paper well written, easy to follow and likely of reasonably broad interest. I only have a few issues with the paper as is;

1) I feel it could be improved with a shortening of paragraphs 2-6 in the discussion. For example, P19 L1-18 could be far more succinct at making the authors point (which I think the authors are overly cautious about).

2) I find the handling of replication difficult to follow. The experiment had 5 treatments with 3 replicates of each, yet it seems like replicates were pooled for data analysis and that ‘replication’ terminology actually refers to selected ROIs? Replication is often a problem with nanoSIMS based studies, but it needs to be clearly outlined as ROIs and true replicates are two very different things. That said, the trends they observe are quite real and their findings are certainly justified.

Few specific comments:

P4 L22-23 – replace ‘ultra-high resolution stable isotopic mapping’ with high resolution secondary ion mass spectrometry

P6 L23 – how realistic are the isotope spikes? Do these values reflect natural conditions?

P9 L14 – did $^{13}\text{C}_2$ create any interference problems for $^{12}\text{C}^{14}\text{N}$?

P9 L23 – how often was a control sample analysed? Presumably a control sample was analysed periodically through the total analysis time to account for instrument fractionation?

P19 L2 – I don’t understand the authors description of 0.2% enrichment detection limit. How do the treatments compare to the control samples? It seems more a question of whether enough data was collected for individual ROIs and how error is then considered. I don’t think the authors need to be so cautious here if they are confident with their method.

Author's Response to Decision Letter for (RSPB-2020-2393.R0)

See Appendix A.

Decision letter (RSPB-2020-2393.R1)

20-Nov-2020

Dear Mr Lyndby

I am pleased to inform you that your manuscript entitled "Amoebocytes facilitate efficient carbon and nitrogen assimilation in the *Cassiopa-Symbiodiniaceae* symbiosis" has been accepted for publication in Proceedings B.

Open Access

You are invited to opt for Open Access, making your freely available to all as soon as it is ready for publication under a CCBY licence. Our article processing charge for Open Access is £1700. Corresponding authors from member institutions

Paper charges

Sincerely,

Reviewer(s)' Comments to Author:

Referee: 1

Comments to the Author(s)

General comments

This paper was really enjoyable to read. This is an interesting and original study investigating the use of dissolved inorganic nitrogen compounds (NO₃⁻ and NH₄⁺) by jellyfish of the genus *Cassiopea* sp. living in symbiosis with dinoflagellates (*Symbiodinium*, formerly clade A). In comparison with other studies published in the literature so far, this one investigates the incorporation of these two compounds at the tissue and single cell level, using the NanoSIMS imaging technique. The methodology is well explained, and the results obtained are convincing. For these reasons, I recommend publication of the manuscript with minor corrections/clarifications.

Reply: We thank the reviewer for the positive response to our manuscript, and hope that all questions and remarks have been answered satisfactorily below.

Methods

Page 6, « Pulse-label incubations » How were the two incubation periods (2h and 6h) chosen? Did the authors perform preliminary tests with different incubation times or do 2h and 6h represent a kind of "standard incubation time"?

Reply: 2 and 6 hours of incubation were chosen based on similar studies previously done with corals. See e.g. Kopp et al. 2015, mBio or Krueger et al. 2018, Scientific Reports. The decision to go with these incubation times was based on the idea that *Cassiopea* represent a system very similar to symbiotic corals.

Page 6, « Pulse-label incubations » Has the ammonium concentration been verified in the sea water in which the jellyfish were preconditioned before incubation in the presence of ¹⁵Nnitrate? Jellyfish could indeed be expected to excrete ammonium, and this would inhibit the incorporation of nitrates (as also suggested by the authors page 18, lines 9-10).

Reply: The nitrogen levels were only monitored in the seawater prior to animal culturing. As such, we cannot exclude an accumulation of ammonium in the water due to excretion by the animal. We now highlight this potential for host ammonium excretion in the discussion as suggested in your point below.

Page 6, Figure 1 : the light intensity used for incubation could be mention here

Reply: Thank you for this suggestion. We have added the photon irradiance used for light incubations in the figure caption.

Results

Page 12, Figure 3: If possible, it would be useful for a non-specialist reader if the authors emphasize the enrichment in symbionts, amoebocytes and epidermis using arrows (as in Figure 1).

Reply: Thank you for this suggestion. We have now added arrows to indicate symbionts, amoebocytes, and epidermis in panel a, d, and g of the revised Figure 3.

Page 13, line 7: the authors found that “specifically, *Symbiodinium* ¹³C enrichment increased 2-fold”. However the LMM gave non-significant p-value at the 95% CI level in this case, is it correct (p=0,127).

Reply: The reported p-value is correct, as the linear mixed model takes into accounts biological replication as well as cellular replication. We have now specifically highlighted the absence of significance in this case in the manuscript.

Page 14, lines 4-6: “Regardless of incubation time and light conditions, medusae incubated with $^{15}\text{NO}_3^-$ did not exhibit any detectable ^{15}N enrichments in any of the holobiont compartments” See previous remark (Methods): See previous remark (Methods): Since ammonium is the main form of inorganic N excreted by jellyfish, this could inhibit the absorption of nitrate. This issue could be slightly developed in the Discussion. For example, in scleratinian corals, nitrate uptake rates are found to be significantly lower in presence of high NH_4^+ concentrations in seawater (e.g. Grover et al., *Limnol & Oceanogr.* 2003).

Reply: You are absolutely right. The lack of nitrate uptake could be attributed to a high *in hospite* ammonium availability for algal symbionts.

We have now expanded on this in the discussion, as suggested (P18-19, L21-4):

“In this context, Grover et al. [51] previously showed that reduced uptake of NO_3^- by algal symbionts can occur in the presence of elevated NH_4^+ concentrations. Hence, the absence of NO_3^- assimilation in the present study could be the result of high *in hospite* NH_4^+ availability for algal symbionts due to the excretion of waste products by the metabolism of the cnidarian host. Reduced or absent NO_3^- uptake by algal symbionts could indirectly benefit the cnidarian host as previous studies have linked high rates of NO_3^- assimilation to reduced rates of carbon translocation from algal symbionts, and increased bleaching susceptibility of corals during heat stress [52, 53].”

Page 14, Figure 5: The non-overlapping error bars (indicating \pm SE) could be a bit confusing, because the authors found no significant differences between treatments with or without light (possibly add/overlay measurement points on each bar ?)

Reply: Good point. Due to the high amount of replication on the cellular level, we felt that the clarity of the figure may be lost if all raw data are included. To give the reader a better idea of underlying raw data we have now included a new Supplementary figure 3 showing boxplots of data distribution as well as individual data points:

Supplementary figure 3 | Boxplot and data distribution of observed ^{15}N enrichment levels in dinoflagellate symbionts and *Cassiopea* sp. host tissue for specimens incubated with ^{15}N -

nitrate (treatment *i* and *ii*) and ^{15}N -ammonium (treatment *iv* and *v*) 6-hours in light (treatment *ii* and *iv*) and 6-hours in darkness (treatment *iii* and *v*). The box in plots show the middle 50% interquartile range (IQR) of data with indicated median, and whiskers the range within 1.5 interquartile range ($Q1-1.5*IQR$ or $Q3+1.5*IQR$).

Page 17, lines 3-4: Note that McCloskey et al. (Mar Biol., 1994) also found that carbon fixed photosynthetically by symbionts satisfy up to 100% of the host's (*Mastigias papua*) daily metabolic carbon demand?

Reply: This is correct. We focus on *Cassiopea*, however we can see why including similar symbiotic jellyfish would be of relevance. This section has now been changed to include symbiotic jellyfish in a broader sense, and McCloskey et al. 1994 added to the references, and reads (P17, L12-15):

“However, our observations are consistent with previous studies on metabolic interactions between symbiotic jellyfish (such as *Cassiopea*) and their symbiont population, which indicate that these jellyfish can produce more than 100% of their daily carbon demand via autotrophic carbon assimilation [45, 46].”

Referee: 2

Comments to the Author(s)

The authors provide novel insights into the symbiosis formed between the ‘upside down jellyfish’ (*Cassiopea*) and *Symbodinium* sp. – a potential model system analogous to corals. Interestingly, in contrast to the recent ‘coral crisis’, *Cassiopea* have thrived, suggesting comparison of the symbioses formed by the related organisms could lead to important applied outcomes. The elegant and well thought out experiment presented here represents the evolution from previous bulk scale isotope tracer studies on the same organisms (i.e., Freeman et al. 2016). I found the paper well written, easy to follow and likely of reasonably broad interest. I only have a few issues with the paper as is;

- 1) I feel it could be improved with a shortening of paragraphs 2-6 in the discussion. For example, P19 L1-18 could be far more succinct at making the authors point (which I think the authors are overly cautious about).
- 2) I find the handling of replication difficult to follow. The experiment had 5 treatments with 3 replicates of each, yet it seems like replicates were pooled for data analysis and that ‘replication’ terminology actually refers to selected ROIs? Replication is often a problem with nanoSIMS based studies, but it needs to be clearly outlined as ROIs and true replicates are two very different things. That said, the trends they observe are quite real and their findings are certainly justified.

Reply: We thank the reviewer for the positive reply and suggestions for improving the description of the results and the discussion throughout the manuscript. To answer the two issues raised by the reviewer here:

- 1) The reviewer suggests to shorten the discussion. However, we believe a thorough and broad discussion that includes examples of previous studies using bulk-analyses on *Cassiopea*, as well as previous NanoSIMS studies on symbiotic corals, is needed to place our study into the proper perspective.
- 2) We agree with the reviewer that replication is an issue in NanoSIMS analyses because replication can happen on the specimen level (individual *Cassiopea*) as well as the cellular level (individual regions of interest, such as symbiont or host cells). As we think that both levels are important to consider for robust statistical analyses, we opted for a linear mixed model that takes into account the nested design of specimen and cellular replication. To clarify this point, legends of figure 4 and 5 have been adjusted to highlight that that plotted

data reflects the mean across cellular replicates across specimen replicates. Further, the number of total replicates from each treatment/tissue area can be found in the supplementary material in Table S1, and this information has now also been added to the “Statistical analyses”-section in the revised manuscript (P10, L1-2).

Few specific comments:

P4 L22-23 – replace ‘ultra-high resolution stable isotopic mapping’ with high resolution secondary ion mass spectrometry

Reply: Changed as requested.

P6 L23 – how realistic are the isotope spikes? Do these values reflect natural conditions?

Reply: The chosen levels are high, but not unnatural, see e.g. Zhang and Fischer 2014, Environ. Sci. Technol. and Fiore et al. 2013, PLoS ONE, for natural values measured in waters around Cuba from which the jellyfish originate. This information and references have now been added to the paragraph (P6, L2-4):

“The experimental concentrations of bicarbonate (3 mM) and nitrate or ammonia (3 μ M) are high, but not uncommon for waters around Cuba where the animals originate [37, 38].”

P9 L14 – did $^{13}\text{C}_2$ create any interference problems for $^{12}\text{C}^{14}\text{N}$?

Reply: Outside of intracellular compartments with very low N content (such as lipid droplets), which do not contribute to average N-isotopic ratios presented here, the count rates of $^{13}\text{C}_2$ were never high enough to significantly enhance the $^{12}\text{C}^{14}\text{N}$ count rate and affect the measured $^{15}\text{N}/^{14}\text{N}$ ratios.

P9 L23 – how often was a control sample analysed? Presumably a control sample was analysed periodically through the total analysis time to account for instrument fractionation?

Reply: Control samples with natural isotopic composition were analysed at least twice per day to monitor for any instrument drift. However, these control isotope ratios were within 20‰ of each other and no corrections were necessary. This information has now been added to the end of this paragraph in the revised manuscript (P9, L9-11):

“Control samples with natural isotopic composition were analysed at least twice per day to monitor instrumental drift. However, these control isotope ratios were within 20‰ of each other and no corrections were necessary.”

P19 L2 – I don’t understand the authors description of 0.2% enrichment detection limit. How do the treatments compare to the control samples? It seems more a question of whether enough data was collected for individual ROIs and how error is then considered. I don’t think the authors need to be so cautious here if they are confident with their method.

Reply: We thank the reviewer for bringing this up. In fact, our detection limit is about 20‰. Control samples were prepared for NanoSIMS in a manner identical to that of samples incubated with isotopic enrichments. We obtained our detection limit simply by analysing a number of regions in the controls of similar size and with similar analytical parameters as we did on samples incubated with isotopic enrichments and determine the isotope ratio standard deviation (SD) from these control images. The detection limit was taken as 3 SD (conservative), and corresponds to about 20‰. This had been added to the revised manuscript as (P9, L4-8):

“The limit for detection of isotopic enrichment was obtained by analysing unlabelled (i.e. control) samples and determine the standard deviation among these analyses. The detection limit was then defined as 3 standard deviations above the control isotopic ratio and all tissue areas with isotopic ratios below this limit were considered unlabelled.”